# Extra-Esophageal Presentation of Gastroesophageal Reflux Disease: 2020 Update

**DOI:** 10.3390/jcm9082559

**Published:** 2020-08-07

**Authors:** Marilena Durazzo, Giulia Lupi, Francesca Cicerchia, Arianna Ferro, Federica Barutta, Guglielmo Beccuti, Gabriella Gruden, Rinaldo Pellicano

**Affiliations:** 1Department of Medical Sciences, University of Turin, C.so A.M. Dogliotti 14, 10126 Turin, Italy; giulia.lupi@unito.it (G.L.); fracice7@gmail.com (F.C.); arianna.ferro@unito.it (A.F.); federica.barutta@unito.it (F.B.); guglielmo.beccuti@unito.it (G.B.); gabriella.gruden@unito.it (G.G.); 23th Internal Medicine Unit, Città della Salute e della Scienza Hospital, C.so Bramante 88, 10126 Turin, Italy; 3Unit of Gastroenterology, Città della Salute e della Scienza Hospital, C.so Bramante 88, 10126 Turin, Italy; rinaldo_pellican@hotmail.com

**Keywords:** gastroesophageal reflux, cough, asthma, laryngo-pharyngeal reflux, chest pain, tooth erosions

## Abstract

Gastroesophageal reflux disease (GERD) is defined by the presence of symptoms induced by the reflux of the stomach contents into the esophagus. Although clinical manifestations of GERD typically involve the esophagus, extra-esophageal manifestations are widespread and less known. In this review, we discuss extra-esophageal manifestations of GERD, focusing on clinical presentations, diagnosis, and treatment. Common extra-esophageal manifestations of GERD include chronic cough, asthma, laryngitis, dental erosions, and gingivitis. Extra-esophageal involvement can be present also when classic GERD symptoms are absent, making the diagnosis more challenging. Although available clinical studies are heterogeneous and frequently of low quality, a trial with proton pump inhibitors can be suggested as a first-line diagnostic strategy in case of suspected extra-esophageal manifestations of GERD.

## 1. Introduction

Gastroesophageal reflux disease (GERD) is a common gastrointestinal (GI) condition with a worldwide diffusion and high prevalence in Western countries. The 2006 Montreal consensus defined GERD as a condition that develops when the reflux of the stomach contents causes troublesome symptoms and/or complications into the esophagus [1]. Tissue damage related to GERD range from esophagitis to Barrett’s esophagus and esophageal adenocarcinoma; troublesome symptoms attributable to reflux can be esophageal (heartburn, regurgitation) or extra-esophageal (EE) [2,3,4,5]. GERD can be further classified by the presence of erosions on endoscopic examination (Erosive Reflux Disease [ERD] and Nonerosive Reflux Disease [NERD]) [5].

GERD-related EE manifestations are frequent and represent a diagnostic and therapeutic challenge, being able to involve lungs, upper airways, and mouth, presenting with asthma, laryngitis, chronic cough, dental erosions, and non-cardiac chest pain (Figure 1).

It has been estimated that one-third of patients with GERD may have atypical or EE symptoms [6]: non-cardiac chest pain is the most common complaint (23.1%), followed by pulmonary manifestations (bronchitis—14.0%, asthma—9.3%) and head and neck symptoms (hoarseness—14.8%, globous sensation—7.0%) [7]. In a prospective European study, the prevalence of EE symptoms was 32.8% in patients complaining of heartburn, with a higher proportion in those with ERD (34.9%) than in those with NERD (30.5%) [6]. Chest pain (14.5%), chronic cough (13%), laryngeal disorders (10.4%), and asthma (4.8%) were the commonest disorders associated with GERD [6].

The prevalence of EE disorders in patients not complaining of typical symptoms of GERD is hard to define, due to the increased difficulty of establishing the correct diagnosis. It has been estimated that between 20% and 60% of patients with GERD have head and neck symptoms without any considerable heartburn. Thus, the diagnosis of GERD-related EE manifestations requires a strong collaboration between specialists to exclude alternative causes [8].

Physiologically, the competence of esophageal sphincters (lower and upper) protect the esophageal and laryngeal mucosa from acid refluxate, while the esophago-glottic closure reflex protects the airway. Peristaltic waves perform mechanical clearance by promoting the progression of the bolus through the esophagus: primary peristalsis is a voluntary process that occurs concurrently with swallowing; thus, it is typical of daytime, while secondary peristalsis is involuntary and predominates during the night. Saliva produced during meals neutralizes acids with its content of bicarbonate and plays a chemical clearance during primary peristalsis [9]. When a reflux event happens, esophageal peristalsis pushes the refluxate back in the stomach, while swallowed saliva neutralize acid [9].

The degree and the duration of acid exposure are responsible for the severity of esophageal mucosal injury and GERD-related symptoms, depending from the incompetence of protective mechanisms. Impairment of the esophageal sphincters is the main predisposing condition: upper esophageal sphincter (UES) insufficiency can be diagnosed by esophageal manometry or pH monitoring. Factors associated with EE are the same as those of GERD, either endogenous, as gastric acidity, pepsin, bile, and pancreatic enzymes, or exogenous such as smoke, alcohol, drugs, and hypertonic solutions [10].

Two main mechanisms have been proposed to explain GERD-related EE manifestations: direct damage induced by the aspiration of gastric materials, and indirect damage, which is vagus nerve mediated.

In the hypothesis of a direct stimulus, cough, laryngitis, or asthma exacerbation appear consequently to a tracheal or bronchial aspirate that stimulates the pharynx and larynx. An intact lower esophageal sphincter (LES) and UES protect from gastroesophago-pharyngeal reflux, while high basal UES pressure and the esophago-glottic closure reflex prevent pharyngeal and laryngeal contact with refluxate [10]. The hypothesis of an indirect mechanism is based on the common embryonic origin and vagus innervation of the esophagus and the bronchial tree, considering cough, bronchial spasm, and cardiac-type chest pain induced by the stimulation of the vagal reflex arc from the distal esophageal reflux [10].

A response to the empiric proton pump inhibitor (PPI) therapy (PPI test) would ideally confirm the diagnosis; however, in a meta-analysis, response to PPIs had only sensitivity of 78% and specificity of 54% in the diagnosis of GERD [11]. GERD-related EE manifestations are less responsive to standard therapy with PPIs [12]. Ambulatory 24-h esophageal pH monitoring is indicated in the evaluation of patients’ refractory to a PPI test and when the diagnosis of GERD is uncertain. This diagnostic test is the only capable of assessing the association between refluxates and reflux symptoms, being particularly useful in detecting GERD-related EE manifestations [5].

Upper gastrointestinal endoscopy is recommended when alarm signs are present (e.g., anemia, undesired loss of weight), in cases of no response to PPI treatment (no decrease of GERD symptoms after short PPI treatment, recurrence of EE symptoms besides 3 months of PPI treatment), dysphagia, suspicious of other causes of heartburn (e.g., eosinophilic esophagitis), long-lasting EE symptoms, the presence of GERD complications, the presence of Barrett esophagus, and fundoplication (before and after).

## 2. Pulmonary Manifestations

An association between GERD and respiratory symptoms has been suggested by several epidemiological studies [13], although a causative association has not been demonstrated yet. Here, we discuss the most frequently reported pulmonary manifestations of GERD: chronic cough, asthma, and aspiration pneumonia.

### 2.1. Chronic Cough

Cough is defined as chronic when it persists over 8 weeks; cough of a much longer duration is defined as chronic refractory cough [14]. Common causes of chronic cough are side effects due to commonly used drugs (especially angiotensin converting enzyme [ACE] inhibitors), tracheo-broncomalacia, chronic obstructive pulmonary disease (COPD), bronchiectasis, asthma, obstructive sleep apnea, rhinosinusitis, and GERD [9,10,13]. In non-smoking patients with normal chest X-rays who are not taking ACE inhibitors, chronic cough is determined in 86% of cases by asthma, postnasal drip syndrome (PNDS), and GERD, although often multiple causes co-exist in a single patient [10].

Several studies suggested a significant relationship between chronic cough and GERD, with prevalence rates of 10% to 56%, which is mainly due to referral bias to centers with specialized interest [6]. In a large prospective European study, the PROGERD study, chronic cough could be attributed to GERD in 13% of patients [6]. In a recent systematic review, Irwin et al. identified GERD as the cause of 85% of chronic cough worldwide, especially in Western countries [15]. In Japan, GERD is described as a rare cause of chronic cough, accounting only for the 7.7% of all causes. The lower prevalence of obesity and the less common Western diet are the main factors associated with the rarity of GERD in this country [16].

When GERD causes cough, GI symptoms can be absent up to 75% of the time, making the diagnosis more challenging [3]. Furthermore, cough and GERD are common diseases and often co-exist, but the association does not imply a causative relationship in all cases: Eastburn et al. showed an occurrence by chance in 25% of cases [17]. Temporal association between reflux episodes and cough could help address correctly chronic cough to reflux, although a diagnostic gold standard is lacking [10].

#### 2.1.1. Pathogenesis

The two main theories proposed to explain GERD-related cough are the reflex theory, considering cough consequent to a vagal-mediated esophageal–tracheobronchial reflex induced by reflux, and reflux theory, suggesting a micro aspiration of refluxed gastric material in the tracheobronchial tract as the cause of cough [16,18].

#### 2.1.2. Diagnosis

GERD-induced cough is usually dry, and it is often exacerbated by postural changes, food intake, and phonation. Chronic cough is quite often the only manifestation of GERD [16]. In patients complaining of chronic cough, it is firstly necessary to exclude pulmonary diseases by performing a radiologic investigation, such as chest x-ray or pulmonary computed tomography (CT). Few cases will require a bronchoscopy also for diagnostic or therapeutic reasons.

When GERD causes cough by irritating the larynx, laryngoscopy can demonstrate signs consistent with “reflux laryngitis” (posterior laryngitis with red arytenoids and piled-up inter-arytenoid mucosa). At bronchoscopy, abnormalities consistent with aspiration can be detected, such as subglottic stenosis, hemorrhagic tracheo-bronchitis, and erythema of subsegmental bronchi. Evidence of inflammation and edema of the larynx and lower airways should not be automatically addressed to GERD because these findings can be associated to other causes of cough or to cough itself. If imaging and endoscopy are normal, it can be assumed that GERD causes cough by stimulating an esophageal–bronchial reflex [3]. In the gastric refluxate, there are multiple potential mediators of cough other than acid, so several mechanisms can be proposed [10].

Patients with laryngeal or pulmonary manifestations of GERD usually are firstly visited by pulmonology and otolaryngology specialists, and only upon a second presentation are they generally admitted by gastroenterologists. In such typical situations, upper gastrointestinal endoscopy is rather often ordered.

A normal esophagogastroduodenoscopy (EGD) is a common finding in patients with GERD-induced cough; only a few have esophagitis or Barrett’s epithelium. Hence, a normal EGD does not rule out the presence of GERD or its involvement in pulmonary abnormalities. Therefore, upper endoscopy should not be performed to diagnose GERD-related asthma, chronic cough, or laryngitis. Furthermore, the diagnosis of esophagitis does not confirm the relationship between GERD and potential EE manifestations [8].

While 24-h esophageal pH monitoring can detect only acid reflux episodes, impedance-pH monitoring can also detect non-acid reflux [19]. During impedance-pH monitoring, reflux episodes are detected considering characteristic impedance changes (e.g., progressing variations in intraluminal impedance), while pH data are used to distinguish acid from non-acid refluxes. The temporal association between reflux events detected at the 24-h reflux monitoring and symptoms is defined by symptom index (SI) and symptom-association probability (SAP) [19,20].

Esophageal manometry and the pH monitoring off-PPI can be recommended in patients with cough unresponsive to treatment and who are considered for surgical options [2].

Recently, Burton et al. have suggested the use of scintigraphy with Tc-99m to identify alterations in the esophageal motility and lung aspiration of refluxate [4].

Given the low availability of pH monitoring, its invasiveness, and the common association between chronic cough and GERD, it is frequent to diagnose GERD-related cough with an empiric trial of PPIs. Up to 79% of patients with cough secondary to GERD experienced a resolution of symptoms after PPI therapy, thus confirming the diagnosis [10]. However, American Gastroenterological Association (AGA) Guidelines recommend 24-h pH monitoring before starting a PPI trial in patients with suspected GERD-related EE manifestations and an absence of typical esophageal findings [5].

#### 2.1.3. Treatment

Although there is poor evidence to support this approach, PPIs are the commonest treatment used in the suspect of GERD-induced chronic cough. Several studies have shown an improvement of chronic cough with this treatment; however, a recent randomized controlled trial (RCT) did not show differences between PPIs and placebo [21,22,23]. A possible explanation can be found in the small sample size included and in the type of quality of life (QoL) questionnaires used to address the usefulness of the treatment [23].

A Cochrane systematic review reported insufficient evidence to conclude for PPI efficacy in treating cough associated with GERD, although some beneficial effect was seen in a sub-analysis [24]. Chronic cough had a high response rate to placebo, and this fact interferes with statistical results in clinical studies. Clinicians prescribing PPI drugs should consider their potential side effects, and maintaining treatment should be planned only when demonstrated useful [24].

Chang et al., in a meta-analysis of RCTs comparing PPI drugs versus placebo, evidenced the efficacy of treatment in patients with GERD-associated cough in a subgroup analysis. In the pooled analysis, there was no effect on the main outcomes, although all studies favored PPIs. The number needed to treat (NNT) to achieve cough resolution was 5. The authors evidenced a smaller effect of treatment on cough compared to the results of non-controlled trials, which was probably related to the placebo effect, which is as high as 85%. A limit of this meta-analysis is the lack of data from RCTs including patients with chronic cough without GERD symptoms. Furthermore, in the included studies, there were no consistent data on the efficacy of dietary changes or surgical treatment [21].

In 2006, the American College of Chest Physician (ACCP) Guidelines on Reflux-Cough Syndrome have been published. These guidelines recommend behavioral changes such as weight loss in patients who are overweight, sleeping with head elevated, and meal avoidance three hours before bedtime. PPI treatment is recommended in patients with symptoms of heartburn and regurgitation; in those with cough but no gastroesophageal symptoms, PPIs should not be prescribed alone, although can be considered in association with lifestyle modifications. In the latter case, prescribing PPIs without behavioral changes are not likely to resolve symptoms [25].

While GI symptoms usually resolve after 4–8 weeks of treatment, the literature suggests that improvement in cough may take up to 3 months. Generally, a positive response to PPIs is evident within a few weeks, being the strongest indicator for disease resolution. It is crucial to reassess shortly the patient response to avoid the prolonged use of useless therapies [10].

Some experts recommend twice daily initial dosing of PPI drugs in patients with chronic cough, although several studies suggested the non-superiority of the twice daily regimen versus the once daily regimen [10]. In resistant cases, the addition of a histamine H2 receptor antagonist (H2-blockers) and/or baclofen may be helpful [22].

Anti-reflux surgery (as Niessen’s fundoplication) may have a role in medical resistant reflux-associated chronic cough when there is not a major motility disorder (absent peristalsis, achalasia, distal esophageal spasm, hypercontractility) [2].

### 2.2. Asthma

Asthma is defined by the American Thoracic Society (ATS) as “a condition with a history of discrete attacks of wheezing, coughing or dyspnea and increase in forced expiratory volume in one-second (FEV1) of 20% from baseline after bronchodilator administration or decrease in FEV1 of 20% after methacholine bronchoprovocation” [26].

Gastroesophageal reflux has been proposed as a trigger for asthma, also when clinically silent, and an effective treatment of reflux could improve asthma control [27].

A significant association between asthma and GERD has been shown in epidemiological studies: up to 50% of patients with asthma have associated GERD [6]. However, the prevalence of asthma in patients with GERD is still uncertain: study reports from 30% to 90%, compared to an average of 24% in controls [9]. A large European prospective study (PROGERD) showed that 4.8% of GERD patients may have asthma [6], while a higher prevalence (24–29%) of silent GERD can be found in difficult-to-control asthmatic cases [28]. Broers et al. reported that the average percentage of GERD prevalence in asthmatic patients was 46.54%, based on symptoms alone and 52.70% based on pH-monitoring and endoscopy, whereas in control groups, the prevalence of GERD was 23.59% based on symptoms evaluation [9].

Although a temporal association between asthma and GERD exists, gastroesophageal reflux does not always trigger asthma [29]. According to Avidan et al., half of all coughs and wheezes in asthmatics are associated with esophageal acid reflux, and at 24-h pH monitoring, it is documented that the reflux episodes lead to cough [18]. However, while an occasional episode of cough can rarely bring to reflux, it is more common that the reflux episode that leads to cough [18].

Similarly, to the challenges encountered in the case of chronic cough, the diagnosis of GERD-related asthma is difficult: upper endoscopy, pH impedance, and the PPI test also when positive, do not always demonstrate the association between the diseases. Silent reflux and night reflux are highly prevalent in patients with asthma and respiratory symptoms: during sleep, the usual protective responses are lacking, increasing the damage provoked by the refluxates [30].

An unresolved question is if asthma worsens GERD or GERD exacerbates asthma. In asthmatic patients, many factors can contribute to GERD worsening: cough and increased respiratory effort, lung hyperinflation, with diaphragm contraction and increased pressure gradient across the LES. Asthma medications such as theophylline, *β*-agonists, and corticosteroids may promote reflux. On the contrary, GERD as the underlying cause of asthma should be suspected in patients with adult onset of asthma, no family history, no allergic component, a low response to traditional asthma medications, symptoms onset preceded by heartburn and regurgitation, or with postprandial worsening [10,28].

#### 2.2.1. Pathogenesis

Asthma and chronic cough share the two main theories of association with GERD. In the reflux theory, the micro aspiration of gastric reflux determines a direct damage to pulmonary parenchyma, causing symptoms such as cough and wheezing, and inducing histologic damage, possibly leading to acute lung injury and acute respiratory distress syndrome. In the reflex theory, refluxates stimulate the vagal nerve, leading to bronchoconstriction [10].

Bronchial hyper-responsiveness is typical of asthma and is defined by an abnormal bronchoconstriction induced by various agents. Esophageal reflux exacerbates asthma by inducing bronchial hyper-responsiveness to the micro aspiration of refluxate, esophageal-triggered vagal reflexes, and esophageal-triggered neuroinflammation through the release of cytokines as tachykinins [27].

#### 2.2.2. Treatment

Lifestyle changes, such as elevation of the head of the bed, smoking cessation, and dietary changes (reduction of fat, chocolate, alcohol, citrus, tomato, coffee, and tea intake, avoidance of large meals and of eating three hours before bedtime) are recommended to improve reflux control and could help obtain improved bronchial symptoms, although there are no RCTs to confirm this hypothesis.

Although PPIs demonstrated superiority over H2-blockers to cure esophagitis, the efficacy of the former in treating GERD-related asthma is still matter of debate: some studies reported an improvement of symptoms and lung function with reflux treatment, while others did not demonstrate this effect [10,31,32]. In a Cochrane systematic review of randomized, placebo-controlled trials conducted in asthma patients, six studies investigated the effect of PPIs and five investigated that of H2-blockers. The authors found no clear effect on lung function, airway responsiveness, or asthma symptoms [29]. Although most of the included trials reported at least one significant outcome, there was no consistency in the results: FEV1 increase [33,34], reduction of *β*-agonist use [34,35,36], significant improvement in asthma symptoms [34,36,37], and improvement of nocturnal asthma [35,38,39] after treatment with PPIs were reported by two or three studies each. Interestingly, only one trial evaluated the effect of behavioral changes and one evaluated the outcome of surgical approach. No study reported hospitalizations or emergency room visits resulting from asthma [29]. A meta-analysis, summarizing PPI treatment in asthma patients, concluded that there was a small but significant improvement in morning PEF (Peak Expiratory Flow) rate after PPI therapy, although it was highly probable that this amelioration had minimal clinical significance; no overall improvement in lung function and asthma symptom scores was revealed. This meta-analysis included studies comparing asthmatic patients with and without diagnosis of GERD: both groups showed small but statistically significant improvements in the morning PEF rate with PPI therapy, although a larger benefit was seen in those with GERD. Differences in treatment length or cumulative PPI dosage were not associated with a better morning PEF rate outcome [31].

Controversial results on the effect of PPIs in asthmatic patients arise from differing methodologies, small sample sizes, and an absence of placebo group of published studies. Currently, there is no evidence to recommend PPIs in all asthmatic patients, while patients with nocturnal asthma or nocturnal reflux might have some beneficial effects [32].

The actual recommendations in patients with GERD-related asthma (with or without concomitant esophageal symptoms) consist of an initial empiric trial of once or twice daily PPIs for 2–3 months. In patients responsive to therapy, PPIs should be tapered to the minimal dose necessary to control symptoms. In those unresponsive, testing for reflux by pH testing or impedance–pH monitoring can rule out pathological reflux [10].

In some study, anti-reflux surgery showed some beneficial effect on GERD-related asthma: disease control scores dropped, and the consumption of asthma medication decreased. However, consistent evidence encouraging the routinely use of this approach is lacking, and further investigation should be performed [29].

## 3. Laryngitis

Laryngo-pharyngeal reflux (LPR) is defined by the 2002 Position Statement of the American Academy of Otolaryngology-Head and Neck Surgery as a disorder of retrograde flow of gastric contents into the larynx and hypopharynx [40]. It is a common GERD-related EE manifestation: up to 10–15% of all visits to otolaryngology offices are prompted by manifestations of LPR [20].

GERD can cause a variety of laryngeal symptoms, such as hoarseness, sore or burning throat, pain with swallowing, sensation of a lump in the throat, cough, repetitive throat clearing, excessive phlegm, difficulty swallowing, and voice fatigue. These complaints are non-specific of GERD and LPR, and they can be also caused by allergens, smoke, and various irritant agents [10]. In a large case-control study, patients with esophagitis or esophageal strictures had higher odds ratios (OR) for pharyngitis (OR: 1.60), aphonia (OR: 1.81), and chronic laryngitis (OR: 2.01) compared with controls [12]. Many patients diagnosed with laryngeal reflux do not suffer from the classic symptoms of GERD [19]: heartburn is absent in more than half of the patients with LPR [40]. In the PROGERD study, the prevalence of laryngeal disorders was 10.4%, and it was associated with older age, longer GERD duration, and obesity. Interestingly, smokers had laryngeal disorders less often than non-smokers, which was probably due to a desensitized laryngeal mucosa [6].

Laryngeal manifestations of GERD can be explained by a direct damage induced by the acid–peptic contact in the larynx via esophago-pharyngeal reflux (micro-aspiration theory), or by an indirect acidification of the distal esophagus through vagally mediated reflexes (esophageal–bronchial reflex theory). Both mechanisms lead to chronic throat clearing and coughing, inducing mucosal damage and typical signs and symptoms [10].

Laryngeal mucosa is more susceptible to injury than esophageal mucosa: acid refluxate, contents of acid and pepsin, and biliary reflux cause inflammatory and precancerous laryngeal lesions. Nevertheless, the absence of saliva clearance leads to more serious damage compared to the esophagus [10].

### 3.1. Diagnosis

Laryngoscopic findings of reflux-mediated disease are erythema, edema, lymphoid hyperplasia of the posterior larynx, ulcerations, subglottic or posterior glottic stenosis, vocal cord polyps, granuloma, leucoplakia, and cancer [10,41]. Although frequent in reflux laryngitis, most of them are non-specific. Edema and erythema, which are often used to define reflux-induced laryngitis, lack specificity and are highly operator-dependent parameters [10]: in fact, signs of laryngeal irritation are present in over 80% of healthy controls [5]. Allergy, smoking, and voice abuse are common causes of laryngeal irritation and induce the same alteration of LPR.

The use of ambulatory pH monitoring to diagnose LPR is debatable. Hypopharyngeal and proximal esophageal pH monitoring have sensitivities of 40% and 55%, respectively [10,42]. Although pH-monitoring detects reflux in only 40% of patients showing symptoms of laryngeal dysfunction, impedance monitoring can detect the presence also of weakly acid and alkaline reflux, gas, or liquid refluxate possibly causing laryngeal dysfunction [10,41].

A promising non-invasive test to diagnose reflux, although still controversial in its clinical applications, is the salivary detection of pepsin [43,44]. Pepsin is a proteolytic enzyme secreted in the gastric fundus as pepsinogen and activated in the acidic environment: its identification in non-gastric sites can detect the presence of significant reflux. Methods to measure pepsin levels are still not standardized, with heterogeneous accuracy. Using the Western blot technique for sputum and salivary pepsin samples in patients with EE reflux, Kim et al. reported a sensitivity and specificity of 89% and 68%, respectively, based on the pH monitoring results [45]. A monoclonal antibodies assay has shown in a recent, prospective, blinded study positive and negative predictive values of 87% and 78%, respectively [46].

### 3.2. Treatment

Hanson et al. reported a great response rate to the medical and non-medical treatment of LPR: half of the patients responded to behavioral changes, while 54% of those who failed this approach responded to H2-blockers [47]. PPI therapy is the standard treatment in patients with chronic throat symptoms if GERD is suspected as the underlying cause, although a single-dose PPI treatment has not demonstrated superiority compared to placebo in treating LPR [48]. An empirical trial of double-dose PPIs is recommended as first-line therapy in patients with suspected LPR to aggressively suppress the hypopharyngeal acid reflux [48]. A 2016 meta-analysis of 13 RCTs on patients with LPR showed an improvement in reflux symptoms (measured with the reflux symptoms index [RSI]) with twice-daily treatment for 3–6 months, although a difference in the response rate and effect on the laryngeal mucosa was not observed between PPIs and the placebo [49]. On the other hand, a recent meta-analysis of controlled studies including patients with LPR demonstrated no benefit of PPI therapy [50]. This negative finding can be partially explained by the difficulty of identifying patients with LPR, due to the absence of a specific diagnostic tool. The diagnosis of GERD-related laryngitis is presumed in the presence of symptoms such as throat clearing, cough, globous, and signs as laryngeal edema and erythema, although these are non-specific for reflux. Patients unresponsive to PPI therapy can have either a non-reflux related disease or a functional component. The lack of effect of PPIs in clinical trials can be also explained by the high placebo response rates of approximately 40%.

Empirical PPI therapy for a period of one or two months is a reasonable initial approach in patients without warning symptoms and with a high suspicion of reflux-related laryngeal disease. If symptoms improve, therapy might be prolonged up to 6 months to allow the healing of laryngeal tissue, after which the dose should be tapered to minimal acid suppression, resulting in continued response. In patients unresponsive to PPIs, impedance or pH monitoring can be used to rule out reflux as the cause of laryngeal complaint.

Ren et al. considered a combination of PPIs and prokinetics effective in improving QoL, although it had no significant effect on the symptoms or endoscopic responses of GERD-related EE [51].

Among non-pharmacological treatments of LPR, diet modification appeared to be effective: patients following a low-fat, high-protein, and alkaline diet had higher rates of symptom resolution [52]. However, a recent systematic review concludes that there is insufficient evidence to recommend diet modifications for LPR [53].

## 4. Oral Cavity

Saliva is main defense mechanism from acid exposure present in the oral cavity: the quality and amount of saliva provide protection through acid clearance and neutralization [54]. The amount of saliva produced varies throughout the day, depending on circadian rhythms and stimulation from food: a salivary flow rate of 0.2 mL/min (milliliters per minute) is the lower limit of normal unstimulated whole saliva output, while 0.7 mL/min is the lower limit of stimulated salivary flow [55]. Saliva functions involve the removal of pathogenic bacteria that can destroy tissues and cause dental caries in conditions of poor oral hygiene. The presence of lysozyme, lactoferrin, thiocyanate ions, and antibodies make the saliva an excellent antibacterial, while its neutral pH protects the inorganic material of the teeth.

Salivary flow volume and swallowing function are significantly reduced in patients with GERD [56]. The reduction of saliva amount leads to oral dryness, sometimes evolving to xerostomia [57]. Gengivitis, defined as the inflammation of the periodontal soft tissue, is a possible consequence of saliva reduction. The coexistence of bruxism can exacerbate periodontal disease [56].

### 4.1. Dental Erosion

Dental erosion (DE) is an irreversible loss of dental hard tissue by a chemical process that does not involve bacteria, and it is a known major oral symptom caused by acid reflux in patients with GERD, according to the Montreal Definition and Classification [1]. The median prevalence of DE in patients with GERD range widely, from 5% to 47.5% [54,58], with higher severity compared to healthy subjects [59].

DE are caused by a combination of extrinsic factors, such as demineralizing acidic foods, acidic beverages, and medications, and intrinsic causes of tooth erosion, such as recurrent vomiting or regurgitation of gastric contents [54]. Hydroxyapatite crystals, the main component of dental enamel, are damaged if exposed to a pH lower than 5.5. Gastric refluxate has often a pH lower than 2.0, being able to erode dental tissues, depending on the duration and the number of reflux episodes and the function of protective factors such as saliva [60]. Although both DE and dental caries determine the loss of mineral component of the teeth, the former occurs in plaque-free surfaces, while dental caries depend on the exposure to weak acids from cariogenic plaque [54]. While DE can be caused by acid reflux, dental caries do not appear to be related to GERD [56]. A defensive role of acid reflux has been suggested in preventing the formation of dental caries by inhibiting bacterial growth in the mouth [59]. Under normal circumstances, saliva withdraw acid and buffer the remaining [58]: in GERD patients, swallowing function and salivary flow volume are significantly decreased, suggesting a role in the pathogenesis of DE [58]. Direct contact with acid is considered the main mechanism of injury: the acid reflux lowers the pH of the oral cavity, leading to dissolution of the inorganic material of the teeth (hydroxyapatite crystals in the enamel), and consequently to DE, with an irreversible loss of dental substance. DE predisposes the teeth to friction (flattening of the occlusal surface) and abrasion (wear of the tooth substance), which can lead to tooth loss, aesthetic deterioration, and a change in facial appearance [61].

DE is classified taking into account the number and degree of severity of erosion: grade 0 (absence of erosion), grade 1 (loss of the enamel-like cream colored appearance), grade 2 (loss of the enamel surface features: smooth dull appearance, without dentin exposure), grade 3 (involvement of enamel and dentin), and grade 4 (severe structural involvement with destruction of the tooth) [59]. DE caused by GERD can involve any surfaces of the teeth, although it is more often encountered on the labial (buccal), occlusal, and lingual surfaces: reflux acid attacks first the palatal surfaces of the upper teeth, and later, if the condition continues, other teeth may be affected. The palatal surfaces of upper teeth are highly susceptible to erosion being the first encountered by gastric reflux; they are relatively far from major salivary glands, and the tongue keeps the contact of the refluxate against them [58]. The lower lingual surfaces are less affected, which is likely because there is plenty of saliva coming from the submandibular glands [58].

In children with GERD, primary teeth are affected more than permanent ones, being less mineralized and thinner; therefore, they are are more prone to acid erosion [62].

Given the high prevalence of DE in GERD patients, collaboration between dentists and gastroenterologists should be promoted. Subjects with unexplainable DE should be referred to the gastroenterology to investigate the presence of undiagnosed GERD [59].

### 4.2. Oral Soft Tissue Disorders

Oral soft tissue can be damaged by GERD, too [56]. Association with GERD has been proposed for tonsillitis, mucosal atrophy, erythema of the soft palate and uvula, glossitis, epithelial atrophy, xerostomia, and dysgeusia [63]. Common oral cavity complaints in GERD patients are oral dryness, acid and bitter taste, halitosis, itching and burning, and pharyngeal discomfort [56].

GERD can induce oral mucosa damage, although mucosal changes are not pathognomonic of GERD: oral candidiasis, Sjögren syndrome, drug-related xerostomia, poor oral hygiene, dietary changes, and smoking-induced oral lesions present with similar patterns [57]. Palatal regions are typically damaged by GERD [60].

Although mucosal lesions have been found in patients with reflux disease, the literature does not evidence differences between GERD patients and healthy controls in periodontal lesions [59]. Given their non-specificity, the oral soft tissue disorders are not considered a GERD-related EE manifestation in the 2006 Montreal consensus [1].

### 4.3. Diagnosis

An early diagnosis and suppression of acid reflux through lifestyle changes and medication have been reported to prevent damage to the soft and hard tissues of the oral cavity [62].

This diagnosis is generally made by inspecting the oral cavity by a dentistry or dental hygienist. Assessment of the oropharynx and larynx for signs of GERD may help the clinician to establish a diagnosis and subsequent treatment of patients. Since DE is the predominant oral manifestation of GERD, dental examination plays an important role in the early diagnosis of GERD in otherwise asymptomatic patients [62]. Association with typical or atypical reflux symptoms should support the suspicion of underlying GERD.

Due to the low sensitivity of diagnostic tests such as endoscopy and pH monitoring, and the low specificity of laryngoscopy, response to acid-suppressive therapy is now considered the first diagnostic step in patients suspected of having GERD-related oral symptoms [64].

### 4.4. Treatment

In patients with DE, preventive and therapeutic strategies are important. Recommended strategies to stop the progression of this condition include taking antacids immediately after heartburn or after the sensation of acid reflux in the oropharynx, rinsing the mouth with neutral pH mouthwash or neutral sodium fluoride, avoiding brushing teeth immediately after reflux episodes, applying fluoride gel immediately after reflux, avoiding xerostomic medications, lubricating oral cavity with saliva substitute, or stimulating salivary flow with sugar-free chewing gum [65]. Dietary changes are recommended, too, such as avoiding highly processed acidic foods that are rich in fats and added sugars (sour candies, spicy, salty snacks, carbonated beverages, energy and sport drinks), while minimally processed and fresh acidic foods (fresh fruit, tomatoes, and savory vegetables) can be included in mixed meals [65]. Behavioral modifications include stopping smoking and good oral hygiene.

Current guidelines suggest empirical therapy with PPIs twice daily in patients with suspected GERD-related oral manifestations. There are currently no studies on the effect of anti-reflux surgical therapy on GERD-related DE. In patients with LPR who do not respond to appropriate PPI therapy, studies suggest that surgical fundoplication does not lead to a further improvement of laryngeal outcomes or throat symptoms. Therefore, surgical fundoplication is not recommended in this context, while it may be considered as a second-line therapy in patients responsive to PPI but relapsing to suspension [66].

## 5. Chest Pain

GERD-related chest pain is the most frequent atypical manifestation of GERD [6,7]. Although the Montreal Classification considers non-cardiac chest pain as an esophageal syndrome, we discuss it separately from the commonest symptoms of typical GERD, such as heartburn and regurgitation, given its similarity in diagnosis and treatment with EE manifestation [19,67].

GERD-related chest pain is defined as recurrent episodes of substernal pain radiating to the back, neck, jaw, or arms, which can last from minutes to hours and is due to pathological esophageal acid exposure [68].

When chest pain does not have a cardiological origin, it is defined as non-cardiac chest pain (NCCP). NCCP includes heterogeneous causes of various severity: musculoskeletal, pulmonary (pneumonia, pulmonary embolism, lung cancer, sarcoidosis, pneumothorax and pneumomediastinum, pleural effusions), vascular (aortic disorders, pulmonary hypertension), drug-related, psychological, and GI disorders (Table 1). Of these, the most frequent etiology of NCCP is GERD [68]. Focusing on epidemiology, NCCP affects both sexes equally, although females tend to consult healthcare providers more often than males. With older age, cardiac chest pain is more common, with a subsequent decrease in the prevalence of NCCP. Chest pain is a common presentation to emergency departments [69], although only 25% of individuals who experience this symptom present to a hospital [70].

Beyond GERD (30–60% of cases), other esophageal causes of NCCP are esophageal dysmotility (15–30%) and esophageal hypersensitivity [68,69,71], alone or in combination.

The mechanism by which gastroesophageal reflux causes NCCP remains poorly understood. It is still unclear why esophageal exposure to gastric content in some patients causes heartburn and in others, it causes chest pain. In addition, the same patient can sometimes experience chest pain and heartburn on other occasions [68].

GERD-related chest pain is induced by abnormal exposure of the esophageal mucosa to stomach acid content. Under the physiopathological aspect, chest pain could be triggered by the stimulation of acid-sensitive chemoreceptors, mechanoreceptors, or thermoreceptors of the esophageal mucosa.

Esophageal NCCP may be alleviated by an assumption of high-dose anti-secretory drugs, although in some cases, it can benefit from nitrate treatment, complicating the differential diagnosis with angina pectoris [66]. Esophageal chest pain is often related to meals, although it can be precipitated by emotions and exercise, being harder to distinguish from cardiac chest pain [72]. Risk factors for coronary artery disease (CAD), such as smoking, obesity and diabetes, are also risk factors for esophageal abnormality and GERD, complicating the diagnostic differential [72]. CAD and GERD can also coexist, and their prevalence increases with advanced age. Hence, signs and symptoms of the latter should not be considered mutually exclusive of CAD [69]. Epidemiological data have shown that 50% of patients with coronary disease have suffered from one or more symptoms typical of GERD [69]. On the contrary, one-third to one-half of patients presenting with severe chest pain have no evidence of CAD after invasive examination [68].

Functional chest pain should undergo differential diagnosis with GERD-related chest pain. It has been defined by the ROME IV classification as a retrosternal chest pain or discomfort, without esophageal symptoms and without evidence of GERD, esophageal motor disorders, or eosinophilic esophagitis (EoE) [73] as the cause of symptoms that have occurred for the past 3 months with a frequency of at least once a week [74]. Suspected mechanisms include abnormal mechano-physical properties of the esophagus, esophageal hypersensitivity, autonomic dysregulation, and altered central processing of esophageal stimuli [74,75].

### 5.1. Diagnosis

When a patient complaints of chest pain, it is necessary firstly to exclude the cardiac origin of pain, using highly available tests such as electrocardiogram, echocardiography, troponin dosage, and, considering the pretest probability, more specific exams as single photon emission computed tomography (SPECT), stress echocardiography, and coronary computed tomography. Coronary angiography remains the gold standard, but it is an invasive test, and its use is limited to highly suspicious coronary ischemic pain, especially in people over 40 years old [68]. Once serious cardiac conditions have been excluded, it is crucial to rule out life-threatening conditions other than ischemic heart disease, such as pulmonary embolism, aortic dissection, and pneumothorax (Table 2).

The upper digestive tract, the biliary tree, the thoracic wall, or the pulmonary system should be further investigated in the diagnostic work-up after life-threatening conditions have been ruled out.

In the suspect of GERD-related NCCP, a PPI trial could be used by primary care physicians as the initial diagnostic tool after the exclusion of non-esophageal causes: rabeprazole 20 mg twice daily for two weeks has shown a sensitivity of 81% and specificity of 62% in diagnosing GERD-related NCCP [76]. In a systematic review of the diagnostic accuracy of the PPI test in these patients, sensitivity and specificity were 0.89 and 0.88, respectively [77]. The recommended duration is at least two weeks of treatment, and any PPI can be used, although a high dose is recommended: from 40 to 80 mg daily for omeprazole, 30–90 mg for lansoprazole, and 40 mg for rabeprazole [78]. The PPI test is defined positive if a reduction of 50–75% of symptoms burden is recorded [79].

Endoscopic pathological findings are less frequent in patients with GERD-related chest pain compared to those with typical symptoms of GERD. In fact, hiatal hernia, erosive esophagitis, and Barrett’s esophagus was found in 28.6%, 19.4% and 4.4% of subjects complaining of NCCP, respectively, compared to 44.8%, 27.8%, and 9.1% of patients with typical GERD symptoms [80]. The ASGE guideline recommended EGD in patients with symptoms suggestive of complicated GERD or alarm symptoms, for follow-up of patients with severe esophagitis to rule out underlying Barrett’s esophagus and to screen for Barrett’s esophagus in patients with multiple risk factors [81]. When NCCP diagnosis is uncertain, it is recommended to perform upper endoscopy to diagnose other conditions apart from GERD as eosinophilic esophagitis.

The 24-h pH monitoring permits revealing reflux events by identifying pH reductions, with abnormal findings in 40–50% of the cases [71]. The AGA suggest using together with esophageal pH recording a symptom reflux association scheme to accurately diagnose when the chest pain symptom is due to gastroesophageal reflux [71]. The impact of pH-impedance measurement is relevant in patients who do not have esophagitis and do not respond to anti-secretory therapy. In fact, some patients experience chest pain triggered by non-acid reflux, which is identifiable by impedance measurement but not pH monitoring [71].

Esophageal manometry can be helpful in distinguishing GERD from esophageal motor disorders as achalasia and distal esophageal spasm [68].

### 5.2. Differential Diagnosis

GERD-related chest pain should be distinguished from NCCP induced by esophageal motility, visceral hypersensitivity, and disorders of gut–brain interaction such as functional esophageal chest pain, reflux hypersensitivity, and functional heartburn [74,82].

Esophageal motility disorders present with an increase of amplitude and duration of esophageal contractions, generating pain. Manometry can measure these contractions, identifying pressure changes along the entire esophageal tract. Various motility abnormalities are associated with chest pain: hypertensive LES, non-specific esophageal motor disorder, hypercontractile esophagus, distal esophageal spasm, and achalasia [69]. A temporal correlation between sustained contractions of the esophageal longitudinal muscle and esophageal chest pain has been demonstrated [83,84].

Visceral hypersensitivity is the mechanism proposed to explain esophageal NCCP in cases with normal pH measurement. In these patients, a non-pathological reflux (based on characteristics or duration) triggers painful symptoms, such as heartburn or chest pain. Visceral hypersensitivity increases the perception of stimuli due to neuronal hyperexcitability as peripheral sensitization of esophageal sensory afferents and modulation of afferent neural function at the spinal dorsal root or the central nervous system [71]. Esophageal sensitivity has been studied by instilling hydrochloric acid into the distal esophagus in subjects affected by NCCP and healthy volunteers: all patients with NCCP had a reproduction of their pain during instillation. In addition, after acid exposure, the pain threshold dropped further and for longer in NCCP patients than in healthy subjects, identifying the development of secondary allodynia (harmless visceral stimulus hypersensitivity in normal tissue close to the lesion), although its mechanism remains unclear [69,85]. Hypersensitivity to visceral and somatic pain may also be caused by central sensitization.

In several GI disorders, such as irritable bowel syndrome, an increase of mucosal mast cells (MMCs) has been shown to be associated with symptom generation. Furthermore, esophageal MMC count can be associated to visceral hypersensitivity and esophageal dysmotility [86].

Disorders of Gut–Brain Interaction (DGBI) have been extensively discussed in the Rome IV classification of functional disorders: they are defined as a group of disorders classified by GI symptoms related to any combination of motility disturbances, visceral hypersensitivity, altered mucosal and immune function, gut microbiota, and/or central nervous system processing [74]. Functional esophageal chest pain, functional heartburn, and reflux hypersensitivity are the main esophageal phenotypes of DGBI, and these are characterized by the presence of chronic symptoms attributed to the esophagus without evidence of structural, inflammatory, motor, or metabolic disorders [74]. Criteria must be fulfilled for the past 3 months with symptom onset at least 6 months before diagnosis with a frequency of at least twice a week [82]. In the suspect of GERD-related NCCP, patients should firstly undergo a high-dose PPIs trial; if there is no response, endoscopy with esophageal biopsies should be performed to rule out EoE. Afterwards, pH monitoring and esophageal manometry should therefore be performed to exclude NERD or esophageal dysmotility. Once all these investigations are negative, the symptom can be considered an expression of a functional disorder [87].

Functional chest pain accounts for more than one-third of the patients diagnosed with esophageal related NCCP; esophageal hypersensitivity, with the painful perception from normal stimuli, is the mechanism proposed to explain this condition. Treatment goals include symptoms control and improvement in quality of life, using neuromodulators (as tricyclic anti-depressants, selective serotonin reuptake inhibitors), alternative and complementary medicine, and psychological intervention [88].

Patients with functional heartburn do not respond to PPI trial, have a normal acid exposure and negative symptom–reflux association, while those with reflux hypersensitivity present with normal acid exposure and positive symptom–reflux association [87].

Given the presence of symptoms unrelated to reflux episodes, functional heartburn is primarily treated with neuromodulators, but psychological intervention and complementary and alternative medicine may also play a role; anti-reflux surgery should be avoided [89].

Patients with reflux hypersensitivity have clinical symptoms during reflux episodes with normal esophageal acid exposure; the mainstay of treatment is esophageal neuromodulators, while surgical anti-reflux management can be used in selected cases [90]. Drugs and behavioral modifications to reduce reflux events are always recommended [90].

It should be highlighted that functional heartburn and reflux hypersensitivity can overlap with GERD [87].

### 5.3. Natural Course

Patients with NCCP have good outcome and higher life expectancy than those with cardiac pain. Thus, NCCP does not change the prognosis of patients with GERD [69].

Although the life expectancy of GERD patients with NCCP is not affected, QoL is often impaired by this complaint: most patients report an impairment of functional status, chronic use of drugs (PPIs, cardiac, and psychiatric), repeated hospital admissions, and multiple cardiac and non-cardiac investigations [69]. As a result, the economic impact of NCCP on the healthcare system is higher than it should be. In addition to the cost of multiple clinical and emergency room visits, hospital admissions, and prescribed drugs, indirect costs, such as loss of working days and patient QoL, should be considered [69,91].

Once it is confirmed that the esophagus is the source of pain, patients are less likely to feel disabled and reduce the request of medical evaluation. When GERD is identified as the cause of pain, anti-reflux therapy is started, generally with good outcome.

NCCP is associated with psychological diseases, such as panic disorder, anxiety, and depression, which can cause chest pain independently from GERD or enhance reflux perception [71,92]. Of all the GERD-related EE manifestations, chest pain is the most associated to psychometric abnormalities. NCCP patients with psychological disorders show lower QoL, more frequent chest pain, and lower treatment satisfaction than NCCP patients without psychological co-morbidity [92].

### 5.4. Treatment

The pharmacological treatment of GERD-related chest pain is complex and still under investigation: the cornerstone is represented by PPIs and H2-blockers, with the former considered the main first-line therapy. Patients with diagnosed GERD (endoscopic findings and/or abnormal pH test) improve symptoms in 78–92% of cases with anti-reflux treatment. In contrast, response to PPI treatment in NCCP patients without objective evidence of GERD range between 10% and 14% [93]. Furthermore, the duration of PPI therapy with has yet to be clarified, although a 2–3 month course is generally recommended [68]. On the other hand, a lack of response to PPI trial of 2 weeks should lead to the discontinuation of PPI treatment [91]. In a recent systematic review and meta-analysis, PPI treatment in GERD-related NCCP was more effective than placebo, while results in NCCP patients without GERD were inconsistent [91]. In an uncontrolled trial, 2 weeks of high-dose rabeprazole (40 mg) resulted in symptom improvement in 81% of NCCP patients with GERD, which was statistically significant when compared with non-GERD-related NCCP patients [76]. Today, a full course of treatment with double-dose PPI, over a period of 2 months, is still considered the best initial therapeutic approach for GERD-related NCCP [71].

Laparoscopic Nissen fundoplication is a surgery technique that restores the anti-reflux barrier by reinforcing EGJ basal pressures, repairing hiatal hernias, and it enhances the peristaltic function of the esophagus. Both complete and partial surgical fundoplication have been performed in patients with GERD-related NCCP: 81–96% of those with correlation of symptoms to reflux events had an improvement of symptoms after surgery compared to those without correlation [71]. Surgical fundoplication has been shown to be more effective in patients with typical GERD symptoms associated to NCCP, and in those who responded to PPI therapy, compared to those with atypical manifestations of the disease and low response to PPIs [71]. This effective procedure has some side effects: Esophagogastric junction is significantly altered after surgery, leading to more frequent motility disorders, bolus pressurization, and post-operative dysphagia. Post-operative dysphagia can affect up to 90% of post-fundoplication patients with various severity (graded in four-point Likert-like scale). Laparoscopic Nissen fundoplication is currently the ‘‘gold standard’’ technique for the surgical treatment of GERD, but it is indicated when an optimal dose of PPIs does not control the disease or medical long-term therapy cannot be taken [87,94].

When chest pain is due to esophageal mucosa hypersensitivity, recommended treatment includes visceral pain modulators such as tricyclic antidepressants (TCA), trazodone, adenosine antagonists, serotonin–norepinephrine reuptake inhibitors (SNRI), and selective serotonin reuptake inhibitors (SSRI). Although trials evaluating pain modulators are small and often not placebo controlled, these medications remain the mainstay of esophageal hypersensitivity. Of them, venlafaxine and sertraline have showed the most encouraging data for pain modulation in NCCP patients [68,71,92].

Given the association between NCCP and psychological disorder, cognitive behavior therapy (CBT) has been investigated as a possible intervention. Demiryoguran et al. found that in patients who underwent CBT, there was a significant reduction in the number of days with chest pain, severity of symptoms, psychological distress, reduced activity due to pain, and depressed mood compared to controls. However, further investigations are required before suggesting CBT routinely to treat NCCP [92]. CBT should be also considered in patients with elevated levels of hypervigilance and anxiety.

## 6. Conclusions

The diagnosis of GERD-related EE manifestation is not simple and often of exclusion. EGD plays a marginal role, being more useful if alarm symptoms are present. The 24-h esophageal pH monitoring is of relevance in the diagnostic work-up of EE manifestations. This test allows diagnosing acid reflux events in the esophagus, and when using pH impedance monitoring, refluxates of both acidic and non-acidic material into the esophagus can be identified as well. A PPI test is often used as the first diagnostic step. In atypical cases, diagnostic tools such as laryngoscopy and bronchoscopy may be useful to detect abnormalities associated with reflux damage.

Table 3 shows schematically shown the main diagnostic tools mentioned above.

Lifestyle modifications, such as elevation of the head of the bed, weight reduction, smoking cessation, and dietary changes (reduction of fat, chocolate, alcohol, citrus, tomato, coffee and tea intake, avoidance of large meals and of eating three hours before bedtime) are always recommended, both in typical GERD and in its related EE manifestations. Pharmacological therapy is used in all forms of GERD. This is especially effective in patients with evidence of acid reflux to pH monitoring. H2 blockers are not superior to PPIs but can be used as a valid alternative. In some difficult-to-treat cases, the association between PPI an H2 blockers can be tried. The anti-reflux surgery can be used in cases of NNCP or chronic cough associated with evidence of acid reflux to pH monitoring in patients responsive but dependent from PPI therapy. In NCCP patients, due to esophageal hypersensitivity, visceral pain modulators should also be considered.

## Figures and Tables

**Figure 1 jcm-09-02559-f001:**
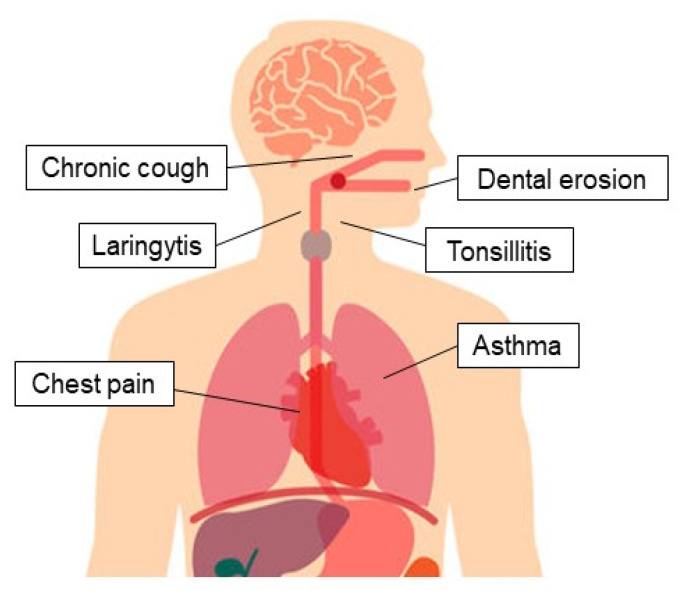
Extra-esophageal presentation of gastroesophageal reflux disease.

**Table 1 jcm-09-02559-t001:** Non-Cardiac Chest Pain Etiologies. GERD: gastroesophageal reflux disease.

Etiological Site	Specific Disorder
Muscoloskeletal	Costochondritis
	Fibromyalgia
Esophageal	GERD
	Esophageal motor disorders (achalasia, hypercontractile esophagus and distal esophageal spasm)
	Esophageal cancer
	Functional chest pain
	Eosinophilic esophagitis
Gastrointestinal	Gastritis
	Pancreatitis
	Cholecystitis
Pulmonary	Pneumonia
	Pulmonary embolism
	Lung cancer
	Sarcoidosis
	Pneumothorax
	Pleural effusion
Vascular	Aortitis
	Aortic dissection
Miscellaneous	Herpes zoster
	Sickle cell crisis
	Psychological disorders

**Table 2 jcm-09-02559-t002:** Life-threatening conditions of chest pain.

Etiological Site	Life-Threatening Condition
Cardiac	STEMI (ST elevation myocardial infarction)
	Cardiac tamponade
	Cardiac wall rupture
Vascular	Aortic dissection
Pulmonary	Pulmonary embolism
	Pneumothorax
	Pneumomediastinum

**Table 3 jcm-09-02559-t003:** Diagnostic tools for extra-esophageal (EE) GERD. EGD: esophagogastroduodenoscopy, PPIs: proton pump inhibitors.

Diagnostic Tool	Recommendation
EGD	Recommended if alarm symptoms (weight loss, age > 50, anemia)
24-h esophageal pH monitoring	Recommended for chronic cough, asthma, laryngitis, oral cavity injury, non-cardiac chest pain, aspiration pneumonia
pH impedance monitoring	Recommended for asthma, laryngitis
PPIs trial	Recommended for chronic cough, asthma, laryngitis aspiration pneumonia, oral cavity injury
Laryngoscopy	Recommended for laryngitis
Bronchoscopy	Recommended for chronic cough

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
