# Peer review of "Extra-Esophageal Presentation of Gastroesophageal Reflux Disease: 2020 Update"

_jcm, 2020, doi:10.3390/jcm9082559_

Round 1
Reviewer 1 Report
This paper reviewed extraesophageal symptoms of GERD with the current evidences. Although it is well written, some parts of the manuscript need to be modified.
Major comments:
- In the introduction, chest pain was firstly regarded as esophageal symptom (page1, line 33). However, it was considered as an extra-esophageal symptom from a statement at page1, line 38 on. According to the Montreal classification, chest pain should be esophageal symptom. The authors need to explain why NCCP is considered as extra-esophageal symptoms.
- Page 7-8, line 277-350. The section as for aspiration pneumonia hardly explain its association with GERD. In my opinion, this section can be omitted. Alternatively, the authors can add some description into a section “2.1.1 pathogenesis of chronic cough” like micro aspiration can cause pneumonia.
- Page 13, line 561-599. It is confusing because the authors discuss mixing together several types of NCCP-causing diseases (functional chest pain, reflux hypersensitivity, functional heartburn, motility disorders, and GERD (with abnormal esophageal exposure)). Pease clarify if GERD-related NCCP includes reflux hypersensitivity and functional heartburn apart from GERD. Concerning the diagnosis, it should be discussed clearly how to distinguish GERD-related NCCP from others and subsequently how to classify functional chest pain, reflux hypersensitivity, functional heartburn, and GERD in GERD-related NCCP.
- Page 15, line 651-689. The treatment should focus on GERD-related NCCP given the title of this manuscript (the treatment for motility disorders should not be discussed here). Also, please discuss the treatment for each GERD-related NCCP phenotype (functional chest pain, reflux hypersensitivity, functional heartburn, and GERD (with abnormal esophageal exposure)).
Minor comments:
- Page 2, line 55, 56. Please explain in more detail as primary and secondary peristalsis contribute to bolus clearance, and saliva delivered by primary peristalsis plays a role of chemical clearance by neutralizing acid with bicarbonate.
- Page2, line 59. Incontinence sounds unsuitable for LES. “Impairment of esophagogastric barrier function” would be better.
- Page 4, line 128. Esophagogastroduodenoscopy is normally abbreviated to EGD.
- Page 4, line 138-139. “Latest pH monitoring instruments” is vague description. “Impedance-pH monitoring” can detect non-acid reflux episodes whereas pH monitoring cannot. Therefore, impedance-pH monitoring should be clearly distinguished from pH monitoring. A paper by Sifrim D et al. (Gut 2004;53:1024-31.) should be cited as a related reference.
- Page 4, line 124. “Esophageal pH monitoring, eventually combined with manometry,” can be taken as simultaneous recording of pH monitoring and manometry. The guideline (ref #2) clearly recommends performing manometry and pH-monitoring separately off-PPI as done for patients with unproven GERD. To clarify the description, it should be modified to “Esophageal manometry and pH-monitoring off-PPI“.
- Page 5, line 195. Reference #25 seems inappropriate for this statement.
- Page 5, line 213. Please add the conclusion of the study (ref #18) saying cough preceding reflux is rare.
- Page 6, line 249-255. Please cite individual studies where appropriate, not one review for all.
- Page 8, line 330, 331. What does “PPI assumption” mean by? Is it meant to be PPI consumption?
- Page 9, line 387. Impedance-pH monitoring cannot detect biliary reflux, but can detect alkaline reflux. The two terms do not mean the same.
- Page 9, line 389-396. Concerning pepsin test for diagnosing GERD, please stress that it is still controversial for clinical application by citing two articles below.
Aliment Pharmacol Ther. 2019 May;49(9):1173-1180. doi: 10.1111/apt.15138.
Clin Gastroenterol Hepatol. 2019 Feb;17(3):563-565. doi: 10.1016/j.cgh.2018.05.016.
- Page 11, line 483-487. Some description including acid and bitter taste and halitosis etc. is repetitive. Please curtail them.
- Page 12, line 541-543. I disagree with including burning sensation in the description of noncardiac chest pain. In clinical practice, we distinguish heartburn from chest pain as much as possible.
- In table 1, NCCP should cover some functional GI disorders such as function heartburn and reflux hypersensitivity defined in Rome IV criteria (Ref #69). Nutcracker has been rarely used since the latest Chicago classification came out (Neurogastroenterol Motil (2015) 27, 160–174). Hypermotility esophageal disorders such as hypercontractile esophagus and distal esophageal spasm can cause NCCP. Esophageal cancer can also manifest itself as chest pain.
- Page 13, line 579-581. Again, nutcracker is no longer used. The latest Chicago classification classified nutcracker as normal.
- Page 14, line 627-629. Central sensitization can be occurred at the brain level as well. Recently hypervigilance is reported to influence hypersensitivity mutually (Aliment Pharmacol Ther. 2018 May;47(9):1270-1277. Am J Gastroenterol. 2020 Mar;115(3):367-375.).
Author Response
We thank the Reviewer for the comments and suggestions which give us the opportunity to improve our manuscript.
Please find enclosed the revised version of our article, with changes tracked according to requirements of the Journal and a letter with the point by point reply.

Reviewer 2 Report
Extra-esophageal presentation of gastro-esophageal reflux disease. Update 2020
This is a very well described and organized the review. However some addition, and explanation are necessary.
Page 1.41- 47. In GEDG and NERD: percent of EE is quite similar in both entities
Head and neck symptoms: it should be in quotation marks
Page 2. It should be noted and underline that patients with laryngeal/ pulmonary symptoms usually firs are under investigation by specialist of pulmonology and otolaryngology. Finally if not would be presented clear diagnosis, those patients are admitted by gastroenterologist. In such typical situation upper endoscopy, not always, but rather often is ordered. Endoscopy would be performed, if no decrease of GERD symptoms after test ( short treatment) with PPI., or recurrence of extra -esophageal symptoms besides 3 months of PPI treatment . Please present briefly indication to upper endoscopy (not only alarm symptoms) as: dysphagia, suspicious of other causes of heartburn (if exist) for example eosinophilic esophagitis, long lasting GERD of extra -esophageal symtoms/comlications, Barrett oesophagus (every 3 years), before and after fundoplication, etc.
Page 2. 50- 54. Please describe briefly how to evaluate UES, and diagnose of UES insufficiency
Page 3-4, 117 – 124. If pulmonary manifestations occur should be done x- ray and CT of chest before eventually ordered bronchoscopy.
If indication to laparoscopic fundoplication is necessary, it could be added two references (please see below), and briefly described advantages and disadvantages of this surgical procedure as well as post-operative dysphagia grading in four-point Likert scale.
1.Marjoux S, Roman S, Juget-Pietu F, et al. Impaired postoperative EGJ relaxation as determinant of post laparoscopic fundoplication dysphagia: a study with high-resolution manometry before and after surgery. Surg Endosc 2012;26:3642-3649.
2. Rerych K, Kurek J, Klimacka – Nawrot E, Błońska- Fajfrowska B, Stadnicki A. High-resolution manometry in patients with gastroesophageal reflux disease before and after fundoplication. J Neurogastroenterol Motil, 2017, 23 (1), 55-63.
Author Response

(The authors gave the same response as above.)
